# Possible Mechanisms of Stiffness Changes Induced by Stiffeners and Softeners in Catch Connective Tissue of Echinoderms

**DOI:** 10.3390/md21030140

**Published:** 2023-02-23

**Authors:** Masaki Tamori, Akira Yamada

**Affiliations:** 1School of Life Science and Technology, W3-42, Tokyo Institute of Technology, O-okayama 2-12-1, Meguro-ku, Tokyo 152-8551, Japan; 2National Institute of Information and Communications Technology, 4-2-1, Nukui-Kitamachi, Koganei, Tokyo 184-8795, Japan

**Keywords:** catch connective tissue, mutable collagenous tissue, echinoderm, sea cucumber, tensilin, softenin

## Abstract

The catch connective, or mutable collagenous, tissue of echinoderms changes its mechanical properties in response to stimulation. The body wall dermis of sea cucumbers is a typical catch connective tissue. The dermis assumes three mechanical states: soft, standard, and stiff. Proteins that change the mechanical properties have been purified from the dermis. Tensilin and the novel stiffening factor are involved in the soft to standard and standard to stiff transitions, respectively. Softenin softens the dermis in the standard state. Tensilin and softenin work directly on the extracellular matrix (ECM). This review summarizes the current knowledge regarding such stiffeners and softeners. Attention is also given to the genes of tensilin and its related proteins in echinoderms. In addition, we provide information on the morphological changes of the ECM associated with the stiffness change of the dermis. Ultrastructural study suggests that tensilin induces an increase in the cohesive forces with the lateral fusion of collagen subfibrils in the soft to standard transition, that crossbridge formation between fibrils occurs in both the soft to standard and standard to stiff transitions, and that the bond which accompanies water exudation produces the stiff dermis from the standard state.

## 1. Introduction

One of the surprising features of echinoderms is their ability to change the mechanical properties of their connective tissues in response to stimulation. Such connective tissue is called catch connective tissue, or mutable collagenous tissue [1,2,3,4]. Catch connective tissue mainly consists of an extracellular matrix (ECM), such as collagen fibrils, proteoglycans, and microfibrils [1,2,3,4,5,6]. Takahashi discovered catch connective tissue at the spine base of sea urchins, and he named the tissue catch apparatus [7]. Catch connective tissue is found in all classes of echinoderms [1,2,3,4].

The body wall dermis of sea cucumbers is typical catch connective tissue. When a sea cucumber, *Stichopus chloronotus*, is mechanically stimulated, its dermis stiffens. When the animal is strongly stimulated, the dermis ‘melts’ into slime [8]. Stiffness changes in the isolated fresh dermis of sea cucumbers can be induced by changing the ionic environment of the external media. The removal of Ca^2+^ in artificial sea water (ASW) softens the dermis, and elevation of the concentration of K^+^ in ASW stiffens the dermis [9,10,11,12,13,14,15,16].

The three-state model, based on detailed dynamic mechanical tests, well explains the mechanical state of the sea cucumber dermis [15]. The dermis in Ca^2+^-free ASW is in the soft state, the dermis in ASW with a normal ionic composition (nASW) is in the standard state, and the dermis in ASW with an elevated concentration of K^+^ is in the stiff state. The results of the mechanical tests suggest that the mechanical parameters of the standard state are not the simple intermediate values between those in the soft and stiff states. Several endogenous substances involved in the stiffness change of catch connective tissue have been identified. Among these, stiffeners and softeners which directly act on the ECM of the tissue are important for understanding the mechanism of stiffness change. Examples of such molecules are tensilins and softenin isolated from the dermis of sea cucumbers. Tensilins are the stiffeners [17,18,19], and softenin is a softener obtained from *Stichopus chloronotus* [20]. In this review, we summarize the information concerning such molecules and discuss their roles in terms of the three-state model. Although several previous papers have reviewed the general biology of catch connective tissue [1,2,3,4], it seems worth focusing on these key molecules. We also provide the information on candidate molecules that possibly act on the ECM. It has been hypothesized that Ca^2+^ would work as a stiffener by crosslinking the ECM components, such as collagen fibrils, of catch connective tissue [1,2]. However, this hypothesis is not generally accepted [3,11], although a type of cell called a vacuole cell, which secretes Ca^2+^ in response to the elevated concentration of K^+^, is present in the dermis of *Stichopus chloronotus* [21]. It is also known that endogenous classical neurotransmitters and neuropeptides affect the stiffness of the catch connective tissue [1,3,7,8,22], but we will not mention these substances in the following sections.

This review also discusses the possible mechanisms of the stiffness changes based on the ultrastructural studies of the sea cucumber dermis in the different mechanical states.

## 2. Tensilin

### 2.1. Tensilins Purified from the Dermis of Sea Cucumbers

Tensilins are stiffening proteins purified from the dermis of sea cucumbers. Tensilin, first partially purified from a sea cucumber, *Cucumaria frondosa*, has an apparent molecular mass of 33 kDa [13]. A tissue-bending test revealed that tensilin, which was at that time called stiffening factor, stiffens the soft dermis in Ca^2+^-free ASW. Later, the stiffening protein was purified, and was named tensilin [17]. Tensilin binds to collagen fibrils and causes the aggregation of collagen fibrils isolated from the sea cucumber in Ca^2+^-free ASW. A full-length cDNA for tensilin was obtained (Table 1).

Another tensilin (*H*-tensilin) was purified from the dermis of a different sea cucumber, *Holothuria leucospilota* [18]. The apparent molecular mass of *H*-tensilin is 34 kDa. *H*-tensilin aggregated the isolated collagen fibrils in a buffer solution, with or without Ca^2+^. Using *H*-tensilin, the mechanical properties of the sea cucumber dermis were evaluated by a quantitative dynamic mechanical test, while the previous studies on the tensilin of *Cucumaria frondosa* used qualitative tissue-bending tests [13,17]. The dynamic mechanical test described the stiffness and the energy dissipation ratio of the tissue. The application of *H*-tensilin stiffened the soft dermis in Ca^2+^-free ASW and decreased the energy dissipation ratio of the dermis (Table 1). However, the application of *H*-tensilin neither stiffened the dermis in nASW, in most cases, nor changed the energy dissipation ratio. The stiffest dermis was induced by the application of ASW with the elevated concentration of K^+^. The energy dissipation ratio of the dermis in nASW was reduced by the elevation of K^+^. The study suggests that *H*-tensilin converts the dermis from the soft to the standard state, supporting the three-state model [15] (Figure 1).

The aggregation of collagen fibrils induced by tensilin strongly suggested that tensilin stiffens the dermis by directly acting on the ECM [17,18]. However, the possibility that the stiffening activity of tensilin was due to its effect on the cells in the dermis could not be completely ruled out. This was because the studies mentioned above used the isolated fresh dermis, which contained living cells [17,18]. A study using the dermal preparations with disrupted cells ruled out this possibility [20]. In this study, the dermis of *Holothuria leucospilota* was first treated with the detergent Triton X-100 to disrupt the cells. The Triton-treated dermis from *Stichopus chloronotus* was also used in the mechanical test (see Section 4), but here, we only describe part of the results obtained from the dermal preparations of *Holothuria leucospilota*. The Triton-treated dermis corresponds mechanically to the standard state of the dermis, and tensilin did not stiffen the Triton-treated dermis. The preparation, which was softer than the Triton-treated dermis, was made by applying repetitive freeze-thaw cycles to the Triton-treated dermis. Such a preparation is called the Triton-FT dermis. The Triton-FT dermis corresponds mechanically to the soft state of the dermis, and the application of tensilin in nASW to the Triton-FT dermis increased its stiffness [20]. The cloning of *H*-tensilin is underway.

### 2.2. Genes of Tensilin and Related Protein from Sea Cucumbers and Other Echinoderms

As was stated above, a full-length cDNA for tensilin of *Cucumaria frondosa* was cloned [17]. The deduced peptide sequence showed that tensilin has an N-terminal domain which is homologous to the tissue inhibitor of metalloproteinases (TIMPs).

Analysis of the *Strongylocentrotus purpuratus* (sea urchin) genome revealed that there are 10 genes homologous to TIMPs (or tensilin) [23]. The effect of a recombinant protein, which is very similar to 1 of the 10 proteins, on the stiffness of the catch connective tissue of sea urchins was reported [24,25]. We will summarize the study using the recombinant protein in the next section. Nothing is known about the stiffening activity of the other 9 proteins.

A transcriptomic analysis of TIMPs using 41 species shows the presence of TIMP genes in all classes of echinoderms [26]. The analysis showed the presence of TIMP genes from 37 species, each of which contains 2 to 45 genes.

Another transcriptomic analysis using the sea cucumber *Cladolabes schmeltzii* shows the presence of 13 TIMP-like genes in its body wall [27]. The proteins of the ECM, as well its modifying proteins, were analyzed. A phylogenetic tree of the TIMPs of echinoderms, created by using the data on *Cladolabes schmeltzii*, and other data which include the analysis mentioned above [26], shows that 2 of 13 transcripts of *Cladolabes schmeltzii* encode proteins close to the tensilin of *Cucumaria frondosa* (Table 1) [27]. A total of 6 other sea cucumber proteins are close to the tensilin of *Cucumaria frondosa*. The authors called these 9 proteins tensilins. According to the authors, tensilins include a protein from *Apostichopus japonicus* (NCBI accession number PIK52999) [28] and one from the Cuvierian tubules of *Holothuria forskali* (NCBI accession number AQR59058), which we will mention below [29]. Later, the TIMPs of the echinoderms were classified into 5 groups [30]. The analysis showed that tensilins belong to group V of TIMPs. The analysis also suggested the possibility that some TIMPs of echinoderms, including tensilin of *Cucumaria frondosa*, inhibit matrix metalloproteinases (MMPs) [17]. One tensilin expressing in the regenerating gut of a sea cucumber, *Eupentacta faudatrix* (*Ef*-tensilin3), is a possible inhibitor of MMP16 [31] (Table 1). Whether *Ef*-tensilin3 is a stiffening protein or not is still unknown. It is also suggested that a TIMP called AjTIMP1 from *Apostichopus japonicus* inhibits MMP1 [32]. Other echinoderm TIMPs, such as a TIMP from *Strongylocentrotus purpuratus* and one from *Acanthaster planci* (crown-of thorns starfish), may have similar functions [32]. It is possible that some TIMPs of echinoderms are neither stiffening proteins nor inhibitors of MMPs. Some TIMPs of echinoderms are assumed to have metalloproteinase-independent activities, as suggested in mammalian TIMPs [30].

**Table 1 marinedrugs-21-00140-t001:** Examples of tensilins of sea cucumbers.

Species	Body Parts (Protein or mRNA)	* Stiffening	Note	NCBI Accession	Reference
*Cucumaria frondosa*	Inner dermis (protein and mRNA)	+		AAK61535	[17]
*Holothuria leucospilota*	Dermis (protein)	+		Not yet cloned	[18]
*Apostichopus japonicus*	Not shown (mRNA)	ND		ALD83456	[33]
*Holothuria forskali*	Cuvierian tubules (mRNA)	ND	Possible stiffener	AQR59058	[29]
	Dermis (mRNA)	+ (** Recombinant)		MZ561455	[19]
*Cladolabes schmeltzii*	Body wall (mRNA)	ND		GFWR01009215	[27]
		ND		GFWR01009682	
*Eupentacta faudatrix*	Regenerating gut (mRNA)	ND	Possible inhibitor of MMP16 (tensilin3)	GHCL01023186	[31]

ND: not determined; * Stiffening: stiffening activity on catch connective tissue; (+): stiffening effect; ** Recombinant: recombinant protein was used in the mechanical test.

The expression of tensilin-like genes from the body parts of sea cucumbers, other than the body wall and the regenerating gut, is also known. An investigation reported 6 transcripts of tensilin-like genes in the Cuvierian tubules of *Holothuria forskali* [29]. The authors compared the protein sequence to the most abundant transcript with those of tensilin from *Cucumaria frondosa* and putative tensilin from *Apostichopus japonicus* (NCBI accession number ALD83456) [33] (Table 1). For *Apostichopus japonicus*, the NCBI accession number ALD83456 is different from the one mentioned above (NCBI accession number PIK52999), but the protein sequences submitted by two different groups are almost identical. Three amino acid residues are different in these two tensilins. The protein from *Holothuria forskali* is highly homologous to tensilins, and this protein was regarded as the tensilin of the species (NCBI accession number AQR59058) (Table 1). Immunohistochemical study showed the presence of tensilin-containing cells in the connective tissue of Cuvierian tubules using light microscopy. Tensilin-like immunoreactivity was found in the elongated cells. The elongated cells seem to correspond to type 1 neurosecretory-like cells, which were revealed by electron microscopy. The type 1 neurosecretory cell is possibly a juxtaligamental cell (see Section 5.2). The study suggested that the tensilin of the Cuvierian tubules is a stiffening protein, and that the connective tissue of the Cuvierian tubules is a type of catch connective tissue.

Several tensilin-like genes are also found from the body wall dermis of *Holothuria forskali* [19]. As was also described above, a species of sea cucumbers sometimes has two or more tensilins. This may be partially due to tissue-specific tensilin variants [19]. One of the seven proteins (Hf-(D)tensilin) (NCBI accession number MZ561455) was selected for further analysis. Hf-(D)tensilin from the dermis is similar to, but slightly different from, the tensilin of the Cuvierian tubules of the same species (Table 1). Hf-(D)tensilin is also similar to tensilin from *Cucumaria frondosa* and tensilin from *Apostichopus japonicus* (NCBI accession number ALD83456) [33]. The authors studied the stiffening activity of recombinant Hf-(D)tensilin produced in *Escherichia coli* (see the next section).

### 2.3. Recombinant Tensilin and Related Protein

In addition to purified stiffening proteins from catch connective tissue, active recombinant stiffening proteins are also useful for understanding the molecular mechanism of its increase in stiffness. In one study, a mechanical test was performed to see if recombinant Hf-(D)tensilin stiffens the fresh dermis of *Holothuria forskali* [19]. In the mechanical test, a dermal section in Ca^2+^-free ASW was stretched at a constant speed, and the tensile force was recorded. The test showed that recombinant Hf-(D)tensilin stiffens the dermis of the sea cucumber (Table 1) similar to the action of the tensilins purified from *Cucumaria frondosa* and *Holothuria leucospilota* [17,18].

Recombinant Hf-(D)tensilin is also useful in the aggregation assay of collagen fibrils extracted from the sea cucumber dermis [19]. The addition of recombinant Hf-(D)tensilin caused the aggregation of the fibrils. The aggregation never occurred in the presence of 4% heparin. It is possible that sulfates from the proteoglycans attached to the collagen fibril are the binding sites of tensilin. Truncated recombinant tensilins were also used in the aggregation assay. In the truncated tensilins, C-terminal 15 or 40 amino acids were removed. The truncated tensilins did not induce the aggregation of collagen fibrils, although these proteins bound the fibrils. The C-terminal amino acid sequence of Hf-(D)tensilin shows that there is a series of positively charged residues, followed by a series of negatively charged residues. These positively and negatively charged elements are also found in the tensilin of Cuvierian tubules of the same species, as well as tensilins from *Cucumaria frondosa* and *Apostichopus japonicus.* The results suggested that electrostatic interaction of the opposite charges of the C-terminal parts of two or more tensilins causes dimerisation/oligomerization, which is involved in the aggregation of collagen fibrils [19].

The effect of recombinant tensilin-like protein from the sea urchin *Strongylocentrotus purpuratus* (NCBI accession number XM775549.2) [24] was also studied in a mechanical test [25]. The tensilin-like protein was cloned from the peristomial membrane of the sea urchin, although some researchers do not regard any TIMPs of echinoderms other than sea cucumbers as tensilins [27]. The connective tissue of the peristomial membrane of sea urchins is a type of catch connective tissue [34]. The sequence of this tensilin-like protein is very similar to one of the 10 TIMPs reported earlier [23]. One amino acid residue is different between these two proteins. The recombinant tensilin-like protein was produced by in vitro translation, and its effect on the compass depressor ligaments, which are catch connective tissues, of another sea urchin, *Paracentrotus lividus*, was studied in a creep test [25]. The application of the recombinant protein to the anaesthetized ligaments, which had been in the soft condition, in nASW slightly decreased the creep rate. Some proteins grouped in TIMPs of echinoderms may also work as stiffeners similar to the tensilins of sea cucumbers.

## 3. Novel Stiffening Factor

Novel stiffening factor (NSF) is another stiffening protein partially purified from the dermis of the sea cucumber *Holothuria leucospilota* [35]. The stiffening activity of NSF is different from that of *H*-tensilin, which causes the transition from the soft to the standard state [18] (Figure 1). NSF stiffened the fresh dermis in nASW, but it did not stiffen the soft dermis in Ca^2+^-free ASW. NSF also stiffened the dermis that had been converted from the soft to the standard state in Ca^2+^-free ASW by the action of *H*-tensilin. These results suggest that NSF causes the stiffness change from the standard to the stiff state (Figure 1). NSF did not aggregate the isolated collagen fibrils. The estimated molecular mass of NSF was about 2.4 kDa. Whether the stiffening activity of NSF is due to the effect on the cells in the dermis or in the ECM is not known.

## 4. Softenin and Other Softeners

Softenin is a softening protein purified from the ‘melted’ dermis of the sea cucumber *Stichopus chloronotus* [20]. The apparent molecular mass of softenin is 20 kDa. Purified softenin softened the Triton-treated dermis of both *Stichopus chloronotus* and *Holothuria leucospilota*. The results suggest that softenin softens the dermis in the standard state (Figure 1). The effect of softenin on the Triton-FT dermis (see Section 2.1) of *Holothuria leucospilota* was also studied. As was already stated, tensilin stiffened the Triton-FT dermis. When tensilin was washed out by nASW, the Triton-FT dermis gradually softened. The application of softenin caused a rapid decrease in stiffness. Softenin is not likely a simple digestive enzyme because the stiffness recovered, although partially, within one minute upon the removal of the softenin [20]. Softenin dispersed the aggregated collagen fibrils that had been treated with tensilin. These results suggest that softenin softens the dermis by breaking crosslinks among the collagen fibrils. Figure 2 shows the hypothetical interactions of tensilin, softenin, and collagen fibrils in the standard and soft states of the dermis, with the assumption that softenin competes for the binding site of tensilin. We have recently obtained a full-length cDNA of a candidate for softenin (unpublished).

Another softening protein has been partially purified from the dermis of the sea cucumber *Cucumaria frondosa* [13]. The protein is called a plasticizer, with an apparent molecular mass of 15 kDa. The plasticizer has not been further purified.

It is also known that a heptapeptide holokinin isolated from the body wall of *Apostichopus japonicus* softens the dermis of sea cucumbers [22]. It had been considered that softening induced by holokinin is due to the effect on the cells in the dermis because of its structural similarity with bradykinin [36]. However, the identification of the holokinin sequence in a partial protein sequence that is homologous to a 5α type collagen suggested the possibility that holokinin is a collagen-derived bioactive peptide [37]. The molecular mechanism of softening induced by holokinin is unknown.

Although the reaction speed of proteases is relatively slow, the partial involvement of proteases may occur in the softening of catch connective tissue. Echinoderms have many metalloprotease genes [23,27,30], and a metalloprotease was found in the body wall of the sea cucumber *Apostichopus japonicus* [38]. It is possible that a protease that causes degradation of the microfibrils is involved in the extreme softening of the body wall dermis of a sea cucumber *Stichopus badionotus* [39] (see the next section).

## 5. Ultrastructure of the Sea Cucumber Dermis

### 5.1. Extracellular Matrix

To understand the mechanism of stiffness change of the catch connective tissue, morphological studies, in addition to physiological and biochemical studies, are necessary. To see if the structure of the dermis of the sea cucumber *Holothuria leucospilota* changes in combination with the stiffness changes, observations of the soft, standard, and stiff dermis were made using a transmission electron microscope [40]. The ultrastructure of the collogen fibrils isolated from the dermis was also studies. The diameter of the fibrils was slightly larger in the stiffer dermis. The comparison of the aggregated fibrils treated with tensilin and those dispersed by the action of softenin after the aggregation (see Section 4) revealed that the tensilin-treated fibril was thicker than the dispersed fibril. It is suggested that tensilin-dependent stiffening is due to the increase in the cohesive forces, resulting in the lateral fusion of subfibrils.

Transmission electron microscopy also showed that collagen fibrils have crossbridges connecting the adjacent fibrils (Figure 3) [40]. Similar crossbridges were reported in various catch connective tissues of echinoderms, including the sea cucumber dermis [41,42,43,44,45,46]. The number of crossbridges increased as the dermal stiffness increased. It was proposed that the increase in the number of crossbridges occurs in the transition, both from the soft state to the standard state, and from the standard state to the stiff state [40].

It was also found that the distance between the collagen fibrils decreased in the transition from the standard state to the stiff state [40]. Such a decrease was not observed in the transition from the soft state to the standard state. These results are consistent with those from the previous investigation, which shows that water exudes in the transition from the standard state to the stiff state, but not in the transition from the soft state to the standard state [47]. It is suggested that some bonding associated with water exudation causes the stiff state [40]. Similar observations regarding collagen fibril packing [48] and water exudation [49] were made in the compass depressor ligaments of the sea urchin *Paracentrotus lividus*. In these experimental conditions, the softening and stiffening of the ligaments accompanied their slackening and stretching, respectively [48,49]. The distance between collagen fibrils did not change in the transition from the soft to the standard ligament, but it decreased in the transition from the standard to the stiff ligament [48]. It is suggested that the stretching of the ligament induced the shortening of the distance. The authors also suggested the possibility that the reduction in the distance between collagen fibrils facilitates stiffening by creating stronger interfibrillar cohesion [48], although the removal of water from the dermis of the sea cucumber caused only slight stiffening [47]. Water exudation may also be the case in the transition from the standard to the stiff state of the compass depressor ligaments of the sea urchin, as was suggested by Raman spectroscopy [49].

As stated above (see Section 4), the partial involvement of proteases in the softening of catch connective tissue cannot be ruled out. Some sea cucumbers, such as *Stichopus chloronotus* and *Stichopus badionotus* show extreme softening of the dermis [8,26,39]. Ultrastructural studies on the unstimulated and the extremely softened dermis of *Stichopus badionotus* showed that the microfibrils, possibly fibrillin microfibrils [5,6], were not observed in the extremely softened dermis, unlike in the unstimulated dermis [39]. On the other hand, the degradation of collagen fibrils was not evident. Some proteases other than collagenases may be involved in the extreme softening of the dermis.

### 5.2. Tensilin-Containing Cell

During the increase in stiffness of catch connective tissue, stiffening proteins such as tensilin should be secreted from cells in response to stimulation. As stated above, such a cell had been revealed in the Cuvierian tubules of the sea cucumber *Holothuria forskali* by light microscopy [29]. The more detailed study by transmission electron microscopy described the morphology of the cells with tensilin-like immunoreactivity in the dermis of the same species [19]. The study showed that tensilin localizes in the secretory granules of a type of juxtaligamental-like cell. According to the authors, there are 3 types of juxtaligamental-like cells distinguished by the size of the contained granules. The tensilin-containing cell is called the type 2 juxtaligamental-like cell because it contains middle-sized granules. The description of the juxtaligamental cell was first completed in the catch connective tissue of a brittle star, *Ophiocomina nigra* [50]. Juxtaligamental cells are ubiquitous in the catch connective tissue of all classes of echinoderms [1,2,13,43,51,52,53,54,55,56,57]. It has been hypothesized that the juxtaligamental cells secrete the substances that regulate the stiffness of catch connective tissue [3,13]. These results partially prove this hypothesis [19]. Further studies are necessary to understand the function of the other juxtaligamental cells in the dermis of sea cucumbers.

## 6. Conclusions

More than 50 years have passed since the discovery of catch connective tissue. Tensilin, a tissue stiffener, and softenin, a tissue softener, acting directly on the ECM, have been purified from the dermis of sea cucumbers, which is a typical catch connective tissue, but the molecular mechanism of the change in stiffness is not fully understood. The dermis of sea cucumbers seems to be the best material for further study. Among the dermal preparations, the preparations with disrupted cells are useful materials for studying the mechanism of the change in stiffness. Isolated collagen fibrils from the dermis are also useful. Further experiments using recombinant tensilin and softenin, in addition to purified proteins, are expected to advance the understanding of the tissue.

Further morphological studies on the dermis are also necessary. For understanding the mechanism of the change in stiffness, it seems important to obtain more information on the crossbridges connecting the collagen fibrils. More knowledge regarding the cells in the dermis is also required, although it happens that a type of juxtaligamental cell contains tensilin. Determining which cell in the dermis contains softenin, and whether a softenin-containing cell is another juxtaligamental cell, are problems yet to be solved.

## Figures and Tables

**Figure 1 marinedrugs-21-00140-f001:**
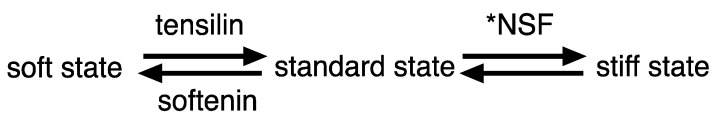
Three-state model of stiffness of the dermis of sea cucumbers. *NSF: novel stiffening factor.

**Figure 2 marinedrugs-21-00140-f002:**
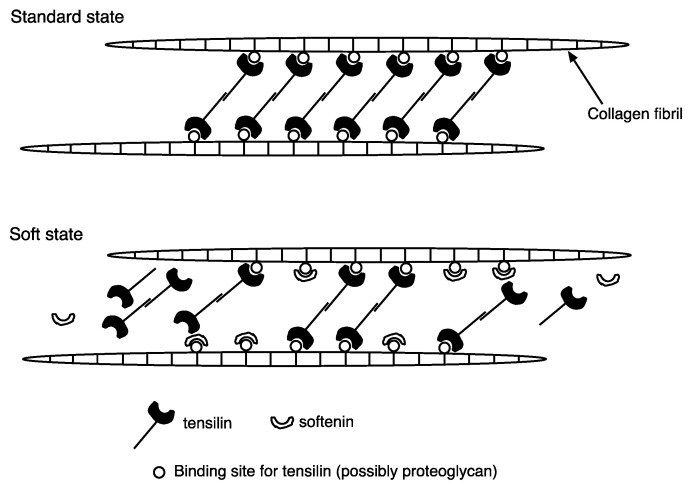
Hypothetical interactions of tensilin, softenin, and collagen fibrils in the standard (**upper**) and the soft (**lower**) states. Tensilin-secreting cells (type2 juxtaligamental-like cells) are not shown.

**Figure 3 marinedrugs-21-00140-f003:**
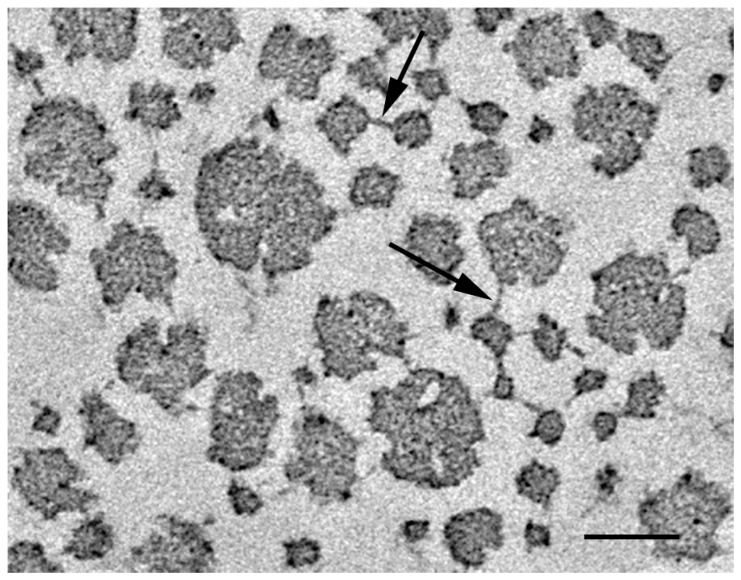
Electron micrograph of the sea cucumber dermis in the standard state. Arrows indicate crossbridges connecting collagen fibrils. Scale bar, 100 µm. Adapted from [40].

## Data Availability

Not applicable. The data presented in this study are available on request from the corresponding author.

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
