# Peer review of "Possible Mechanisms of Stiffness Changes Induced by Stiffeners and Softeners in Catch Connective Tissue of Echinoderms"

_marinedrugs, 2023, doi:10.3390/md21030140_

Round 1

Reviewer 1 Report

The manuscript type is given as “Communication” when it is clearly a review and it is stated that it is a review (lines 51, 61). The manuscript type should be changed to “Review”.

 Line 51 “In this review, we summarize recent knowledge…” Perhaps the word “recent” should be removed since only a quarter of the citations are to work in the last 10 years, and there are only five citations later than 2016. The authors don’t appear to have updated their literature search much since their 2016 paper in PLOSONE. Which is not necessarily a bad thing – provided that this review addresses a specific aspect of interest and if there is important recent work that this is included.                                                                        

There is a recent (2021) publication that is a review with 175 references that the authors might find interesting: The Mutable Collagenous Tissue of Echinoderms: From Biology to Biomedical Applications By I. C. Wilkie ; M. Sugni ; H. S. Gupta ; M. D. Candia Carnevali ; M. R. Elphick DOI: https://doi.org/10.1039/9781839161124-00001 . For the submitted article here, the authors may benefit from clarifying which aspect of this subject they are focussing on in this review, and how that distinguishes it from that recent published comprehensive review on the topic, with reference to that book chapter. A sentence might be all that is needed to place this submission in context.

 Figure 2 should have the citation in the figure caption along with a copyright statement. This image is a modified portion of the image in the cited work so the caption should perhaps also say “adapted from…”

 It is not really the best way to write a review article where paragraphs start with “Dr X found that….”. It is much better to write about the topic and then include the citations within that story. It would really be worth your while to substantially change the way this is written. That book chapter mentioned above is a good example of this preferred way of writing, which is far more useful for the readers. None of the paragraphs (or perhaps even sentences) in that start with “Prof Z et al. ….”. 

 It would be helpful to include a conclusion, summing up the main research ideas provided in the review.

 English: Line212 Softers should be softeners.  Line 51 Knowledges should be knowledge.

Author Response

We appreciate the comments made by the reviewer. We re-wrote the manuscript by changing the way of writing. We also added a recent publication which the reviewer has quated to References. The other responses are as follows.

Point: The manuscript type should be changed to “Review”.

Response: This article is a short review as the reviewer pointed out. But, at submission, we confirmed the Publisher that short review is acceptable as Communication.

Point: Perhaps the word “recent” should be removed.

Response: We removed the word accordingly (Line51).

Point: Figure 2 should have the citation in the figure caption along with a copyright statement. 

Response: We added a copyright statement.

Point: It would be helpful to include a conclusion, summing up the main research ideas provided in the review.

Response: We wrote Conclusions accordingly.

English: Line212 Softers should be softeners.  Line 51 Knowledges should be knowledge.

Response: We corrected the words accordingly.

Reviewer 2 Report

In this manuscript, the authors reviewed recent studies on proteins (softenin and tensilin) responsible for the mechanical states (soft, standard, and stiff) change of sea cucumbers. The experiments were well performed and the data delivered are solid to support the conclusion. The manuscript was reasonably written and easy to understand. It is thus acceptable for publication in the journal of Marine Drugs after minor revisions.

Commnets:

1, information concerning Novel Stiffening Factor (NSF) th in Fig. 1 and page 5, lines 204-211 should be enhanced.

2, a graph summarizing the possible interation between softenin and tensilin with collagen is encouraged. 

Author Response

Comments:

1, information concerning Novel Stiffening Factor (NSF) in Fig. 1 and page 5, lines 204-211 should be enhanced.

2, a graph summarizing the possible interaction between softenin and tensilin with collagen is encouraged. 

Responses:

We appreciate the comments made by the reviewer.

1, We wrote more about Novel Stiffening Factor (NSF) (lines 209-219 in revised manuscript).

2, We added a new Figure (Figure 2 in revised manuscript) showing the possible interaction. 

Round 2

Reviewer 1 Report

The manuscript has been improved appropriately.